# Insights behind the Relationship between Colorectal Cancer and Obesity: Is Visceral Adipose Tissue the Missing Link?

**DOI:** 10.3390/ijms232113128

**Published:** 2022-10-28

**Authors:** Alice Chaplin, Ramon Maria Rodriguez, Juan José Segura-Sampedro, Aina Ochogavía-Seguí, Dora Romaguera, Gwendolyn Barceló-Coblijn

**Affiliations:** 1Institut d’Investigació Sanitària Illes Balears (IdISBa, Health Research Institute of the Balearic Islands), 07120 Palma, Spain; 2Consorcio CIBER, M.P. Fisiopatología de la Obesidad y Nutrición (CIBEROBN), Instituto de Salud Carlos III (ISCIII), 28029 Madrid, Spain; 3General & Digestive Surgery Department, University Hospital Son Espases, 07120 Palma, Spain; 4School of Medicine, University of the Balearic Islands, 07120 Palma, Spain

**Keywords:** obesity, colorectal cancer, visceral adipose tissue, tumor microenvironment, adipocytes

## Abstract

Colorectal cancer (CRC) is a major health problem worldwide, with an estimated 1.9 million new cases and 915,880 deaths in 2020 alone. The etiology of CRC is complex and involves both genetic and lifestyle factors. Obesity is a major risk factor for CRC, and the mechanisms underlying this link are still unclear. However, the generalized inflammatory state of adipose tissue in obesity is thought to play a role in the association between CRC risk and development. Visceral adipose tissue (VAT) is a major source of proinflammatory cytokines and other factors that contribute to the characteristic systemic low-grade inflammation associated with obesity. VAT is also closely associated with the tumor microenvironment (TME), and recent evidence suggests that adipocytes within the TME undergo phenotypic changes that contribute to tumor progression. In this review, we aim to summarize the current evidence linking obesity and CRC, with a focus on the role of VAT in tumor etiology and progression.

## 1. Introduction

Colorectal cancer (CRC) is the third most common cancer in men and the second most commonly occurring in women, as well as the third leading cause of cancer death worldwide [1]. Furthermore, the incidence of CRC is expected to increase by 60% by 2030 [2], particularly in developed countries, where its burden is three to four times higher than in developing countries [3]. The main modifiable risk factors associated with CRC are low physical activity, overweight or obesity, poor dietary habits (including high intake of red and processed meat and low dietary fiber intake), alcohol consumption, and smoking [4], all of which point towards the importance of modifiable lifestyle factors in CRC development. 

Obesity is currently growing at an alarming rate worldwide, reaching epidemic proportions. In 2016, the World Health Organization estimated that 39% of the world’s adult population was overweight, of whom 13% had obesity [5]. Increasing evidence has directly linked obesity with cancer incidence and cancer-associated death, and it is considered a risk factor for at least 13 different types of cancer, including CRC [6,7]. Obese individuals present a 1.3-fold greater relative risk of colon cancer compared to non-obese, particularly seen in men [6,8], and high body weight and excess adipose tissue have a negative impact on the clinical outcomes in approximately 20% of all cancer cases [6]. Furthermore, an increase of 2 cm in waist circumference is associated with a 4% greater risk of CRC [9]. The World Cancer Research Fund and the American Institute for Cancer Research published in 2018 the most recent cancer prevention recommendations based on the latest evidence available [10], stating that maintaining a healthy weight throughout life is key to cancer prevention. Their review of the data published regarding the association between body mass index (BMI), waist circumference, and/or waist-to-hip ratio and CRC was clear: the relationship between greater body fatness and a higher incidence of CRC is convincing.

Although the implementation of screening programs in high-income countries has led to a significant reduction in the incidence of CRC in the population over 50 years of age, the incidence in people under 50 years of age has unexpectedly increased significantly for reasons that are still unclear. The most recent report on cancer facts and statistics in the United States by the American Cancer Society (2020–2022) evidences that CRC patients are getting increasingly younger, shifting from a median age of 72 years for diagnosis in the early 2000s to 66 years currently [11]. This could be partly due to an increase in screening in younger adults, since data from this same cohort indicate that 13.6% of adults aged 40–49 years old reported having a colonoscopy in the past 10 years in 2013, compared with 6.4% in 2000. However, data analysis also shows that the driving factors for the trend observed are individuals between 20–30 years old who do not routinely undergo screening. Furthermore, CRC rates have increased equally for early and advanced-stage cancer, which is not in accordance with a screening effect. Yet, it has been suggested that the rise in CRC in young adults may have been promoted by an increase in excess body fat; however, it has likely been hampered by the decrease in alcohol and smoking among this population [11].

One of the main hypotheses that aims to explain the link between obesity and CRC concerns the role of adipose tissue in its pathogenesis. Adipose tissue is a large endocrine organ that regulates energy and metabolic homeostasis, and its association with cancer is based not only on epidemiological observations but also on the fact that adipocytes are a main component of the tumor microenvironment (TME) for certain cancers, such as breast and gastrointestinal malignant tumors [12,13,14]. Excess adipose tissue promotes the secretion of adipokines, cytokines, and reactive species, which could be one of the key aspects in promoting tumorigenesis, as discussed further in this review [15,16,17]. Furthermore, recent evidence specifically points towards visceral adipose tissue (VAT), which surrounds major internal organs, as the culprit of important health disorders, including cancer [18]. However, despite significant efforts in the field, the exact molecular mechanisms underlying the relationship between obesity and CRC remain unsolved.

The TME is composed of non-malignant cells, vessels, lymphoid organs, and a variety of immune and stromal cell types, including adipocytes, macrophages, fibroblasts, monocytes, neutrophils, immune T-cells, and B-cells [12], and is located either at the centre, margin, or near the tumor itself [19]. In the last decade, emerging evidence has pointed out the central role it plays in the development, growth, and promotion of cancer, having a significant impact on the metabolic features and behaviors of tumors [20]. The definition per se of the TME is under constant renewal, due to the fast-paced and updated publishing of knowledge on its composition and function [19]. Although there are many features of interest regarding the role of TME in CRC etiology and progression, this review aims to tease out the relationship between visceral adipose tissue found in CRC-associated TME and the potential implications of such molecular interactions in CRC treatment strategies.

## 2. Colorectal Cancer: Etiology, Prognosis, and Treatment

CRC generally originates from a polyp in either the colon or rectum, which eventually progresses to CRC over an estimated 10–15 years. There are three main pathways: (1) the adenoma-carcinoma sequence (also known as the chromosomal instability sequence), which occurs in 70–90% of CRC cases [21], and is defined by a series of recurrent driver mutations in *APCR*, *KRAS*, *SMAD4*, and *TP53* genes, as well as in the mismatch repair genes, which accumulate and ultimately lead to CRC; (2) the serrated pathway, which is found in 10–20% of CRC [21], and during which hyperplastic polyps are the precursor lesions to CRC and evolve into invasive adenocarcinomas; and (3) microsatellite instability, representing a minor amount of CRC cases (2–7%), whereby there is either hypermethylation of the *MLH1* gene (in sporadic CRC) or gene mutations occurring in *MLH1* and *MSH2* [22]. Positive family history is thought to be behind 10–20% of patients, and it is estimated that the heritability of CRC ranges from 12% to 35%; however, although some cancer susceptibility has been identified, the causes linked to family history and heritability are still largely unknown. In this sense, only around 5–7% of them are associated with a recognizable syndrome of hereditary CRC [21], including non-polyposis (Lynch syndrome and familial colorectal cancer) and polyposis syndromes.

CRC is associated with nonmodifiable risk factors, such as being male, increasing age, and genetics. Epidemiological findings show that greater BMI is more strongly correlated with an increased CRC risk in men than in women [23,24]. These higher rates of CRC incidence and mortality in men could be due to various biological (genetic) and behavioural factors, as well as to the fact that men present more visceral fat compared to women [25]. In this sense, CRC is largely affected by modifiable lifestyle factors (Figure 1) [26]. These latter ones include physical activity, overweight or obesity, diet (high intake of red and processed meat and low dietary fiber intake), alcohol consumption, and smoking. Low socioeconomic status has also been associated with an increased risk of developing CRC, thought to be due to the fact that most of the modifiable lifestyle behaviours (poor diet, physical inactivity, smoking, and obesity) account for 30–50% of the socioeconomic imbalance regarding CRC risk. In line with this, it has been reported that 47% of CRC cases in the USA and up to 45% of them in the UK have been estimated to be attributable to modifiable risk factors [27]. Furthermore, a recent study carried out in Norway estimated that the adherence to lifestyle recommendations in terms of physical activity, nutrition, and smoking behaviors by 50% of the population would lower cancer mortality by 11% by 2030, vs. a reduction of 7% through screening and 12% through improved treatment [28]. Lastly, other risk factors include gut microbiota composition, inflammatory bowel disease, type 2 diabetes, and drug use [21], which have been extensively reviewed elsewhere [29,30,31]. However, of note, similarities have been described between cancer TME-associated VAT and adipose tissue in type 2 diabetes and obesity, presenting inflammatory features. These include the presence of macrophages and the secretion of cytokines and adipokines, which all together promote an inflammatory state [17]. Furthermore, it has been proposed that the gut microbiota is part of the tumor organismal environment, a recent concept that refers to microenvironments that are distant from the cancer lesions but that can still impact its development [19].

CRC can be managed either following surgical treatment, chemotherapy, or a combination of both. The main treatment for CRC is generally surgical resection of the primary tumor and lymph nodes; the quality of CRC resection is crucial, and laparoscopy has become the standard technique in many countries, with proven short-term benefits [32]. Early-stage cancers could benefit from local treatment via an endoscopy, which is recommended in cases of suspected superficial invasive carcinoma [33]. Lastly, in both colon and rectum cancer, chemotherapy may be used in addition to surgery in certain cases. The decision to add chemotherapy to CRC management depends mainly on the stage of the disease. According to the American Cancer Society, CRC can often be cured if detected at an early stage [27]. On one hand, CRC mortality has decreased by 55% since 1970 due to improved treatments and increased screening programs. From 2014 to 2018, the death rate decreased by nearly 2% each year. On the other hand, deaths from CRC actually increased by 1% from 2008 to 2017 in the population under 55 years old [34], thought to be due in part to an increase in excess adipose tissue and body weight [11]. Considering the fact that the main risk factors for CRC are modifiable, CRC mortality and incidence can actually be reduced by primary prevention strategies (controlling for risk factors associated with lifestyle), secondary prevention mechanisms (prevention and screening), and tertiary prevention (better cancer treatment).

## 3. VAT and Cancer: The Evidence So Far 

Despite all the evidence supporting a direct association between obesity and CRC incidence, the specific molecular mechanisms underlying this link are still unclear; however, the generalized inflammatory state of adipose tissue in obesity is thought to play a role in the association between CRC risk and development. Chronic inflammation is considered one of the major drivers of cancer initiation [35], and the disbalance of inflammatory homeostasis of obese adipose tissue has been explored as a possible molecular link. In this section, we aim to review the role of VAT in tumor etiology and progression in the context of CRC.

There are two main types of adipose tissue: white adipose tissue (WAT), whose main function is energy storage and regulation of systemic metabolic processes, and brown adipose tissue, which specializes in thermogenesis [36,37]. WAT can then be subdivided into two major types: subcutaneous adipose tissue (SAT), which is located under the skin, and visceral WAT, or VAT, which is located in the intra-abdominal cavity and has been widely studied for its association with conditions such as the metabolic syndrome, cardiovascular disease, and cancer, among others [8,13,15,17,18,36]. Most importantly, it seems that inflammation is a common characteristic of all these conditions.

VAT is regarded as an active and dynamic tissue thought to be an important player in the immune response and has been linked to insulin resistance, hyperinsulinemia, hyperglycemia, and oxidative stress [37]. All of these lead to important hormonal changes, including an altered insulin/insulin-like growth factor (IGF) axis and dysregulated levels of adipokines, such as adiponectin and leptin [38], which are, to date, the most studied hormonal systems in this context. Strikingly, it has even been suggested that the amount of VAT and the VAT/SAT index could actually be more indicative of CRC pathogenesis than other measurements such as BMI [35]. Structurally, VAT is highly cellular, vascular, and innervated and is made up of adipocytes, which contain unilocular lipid droplets (making up 95% of the cell volume), and a stromal vascular fraction (SVF) that contains heterogeneous cell populations, including preadipocytes, endothelial cells, pericytes, and immune cells (macrophages, T-cells, neutrophils, and lymphocytes) [35]. Because of its close proximity to many internal organs, it is known to dynamically communicate with them. For example, VAT is close to the portal vein, leading to direct drainage of excess free fatty acids and inflammatory factors, such as cytokines, into the liver, which can lead to hepatic misfunction and injury. Thus, this example highlights the fact that increased lipid delivery to crucial organs and tissues can alter metabolism significantly and contribute to a state of low-grade inflammation, creating a favorable environment for tumor development [37,39]. 

Furthermore, it has been reported that the phenotypical switch of M2 macrophages (anti-inflammatory and adipostatic) to the M1 phenotype (proinflammatory and proadipogenic) occurs when adipose tissue macrophages infiltrate VAT. This leads to an altered M1/M2 balance and creates an inflammatory microenvironment characterized by an increase in proinflammatory cytokines and other factors, which ultimately promote tumor growth (Figure 2) [37].

The immunomodulatory role of VAT has been widely studied in the context of metabolic disorders such as obesity, whereby proinflammatory immune cells such as macrophages, neutrophils, and cytotoxic T cells and B cells are recruited, leading to fibrosis, oxidative stress, and insulin resistance [40]. In a state of excess fat, the adipose tissue undergoes what is known as “adipose tissue remodelling” [41], whereby the number and size of mature adipocytes increases, SVF precursor cells are recruited towards the adipocyte lineage, and hypertrophic adipocytes secrete a range of adipokines and cytokines involved in adipocyte differentiation into mature adipocytes. More specifically, they secrete adipokines (e.g., leptin, adiponectin, resistin, ghrelin) and inflammatory cytokines (e.g., tumor necrosis factor-α (TNF-α), interleukin-1 (IL-1), IL-6, IL-12, and IL-23), among others, all of which are particularly relevant to CRC development, and contribute to the characteristic systemic low-grade inflammation associated with visceral obesity [37]. Furthermore, an increase in adipose tissue leads to inadequate vascular oxygenation of adipocytes, which causes hypoxia and oxidative stress with the resulting overproduction of cytokines and adipokines [37]. Overall, all these factors act on a range of signalling pathways, including phosphoinositide kinase-3 (PI3K)/serine-threonine-protein kinase (AKT) activation, which regulates cell survival and cell growth and thus causes hyperplasia, proliferation, and carcinogenesis in colon cells [37].

Thus, adipocytes are an integral part of the TME, sometimes referred to as cancer-associated adipocytes (CAAs), especially in tumors deeply associated with adipose tissue, such as breast cancer. Therefore, these cells establish a bidirectional crosstalk with tumor cells that could play a critical role in cancer development. In this context, in order to understand the molecular links between colon cancer and obesity, two fundamental questions will be addressed herein: (1) how do VAT adipocytes participate in the inflammatory milieu of the TME and (2) how do they interact with tumor cells to promote tumor progression? 

### 3.1. Role of VAT-Secreted Adipokines and Cytokines in CRC

#### 3.1.1. Adipokines and CRC

The contribution of TME adipocytes to cancer development goes beyond metabolic cooperation and is closely associated with the oncoinflammatory pathways within the TME, whereby adipocytes are characterized by a proinflammatory and protumorigenic adipokine profile [42]. Furthermore, it is thought that adipokines can have protumorigenic effects in the gastrointestinal tract; however, it is still not known whether they act directly on gastrointestinal mucosal cells or in an indirect, paracrine manner by inducing local inflammation [37]. In visceral obesity, a reduction in adipokines with anti-inflammatory and anticarcinogenic properties can lead to an increase in mitogenic signals, decreased cell apoptosis, and increased proangiogenic activity [43]. In the context of CRC, the most relevant adipocyte-secreted hormones are adiponectin, leptin, ghrelin, and resistin [37].

##### Adiponectin

Adiponectin is an insulin-sensitizing hormone secreted by adipocytes which is inversely correlated with insulin resistance and visceral obesity. It has been suggested that adiponectin or its analogues could become useful in the management or chemoprevention of CRC and, together with leptin, is one of the most studied adipokines in the field of CRC. Adiponectin receptors can be found in the colon epithelium, and in vitro data suggest they could be modulating several intracellular signalling pathways regulating cell growth and proliferation [37]. However, data in human studies are still inconclusive and it is not clear whether adiponectin exerts a protective role in CRC. On one hand, an inverse association has been described between plasma adiponectin levels and a higher risk for colonic polyps in prediabetic subjects [44] and in patients with colorectal adenoma [45,46,47]. Furthermore, immunohistochemical analysis of adiponectin receptors in colorectal tissue from newly diagnosed cancer patients found that expression levels were differentially associated with CRC (vs. noncancerous tissue): on one hand, AdipoR1 expression was inversely correlated with nodal stage, whereas AdipoR2 was positively associated with the tumor, node, and metastasis (TNM) stage [48]. On the other hand, two prospective studies showed no association between adiponectin levels and the risk of colon adenoma [49,50], and one study even reported that adiponectin levels correlated with a higher risk of CRC [51]. 

##### Leptin

CAAs produce high levels of leptin, also an insulin-sensitising hormone that regulates adiposity by modulating satiety signals and energy expenditure [52]. The role of leptin in obesity-associated cancers has been previously reviewed [37,53,54]. A range of in vitro, in vivo, and translational studies have shown that leptin could be acting directly on cell proliferation and apoptosis in CRC via the PI3K/AKT/mTOR signalling pathway [54]. As with adiponectin, leptin receptors are overexpressed in colon cancer cells, suggesting an active communication mediated by this hormone [55], whereby leptin is able to stimulate cell DNA synthesis and growth in colon cancer cells [56,57,58], suggesting that when levels are increased, it is capable of acting as a growth factor via the MAPK and PI3-K pathways [57]. Furthermore, the absence of leptin receptor expression in histopathological samples from CRC patients has been associated with decreased tumor proliferation and metastasis and, if associated with obesity, chemotherapy resistance. Furthermore, an imbalanced leptin/adiponectin ratio can impact the JAK/STAT signalling pathway as do inflammatory cytokines, promoting CRC development [54].

##### Ghrelin

Ghrelin is an orexigenic peptide that acts as an important physiological regulator of lipid metabolism [43] and stimulates growth hormone release, adipogenesis, and changes the growth processes of neoplastic tissues, among other functions. It is known as the “hunger hormone” because of its stimulatory effect on food intake. Interestingly, ghrelin may act as an antiapoptotic or proapoptotic factor in different cancer cells [37]. However, data are still inconsistent (reviewed extensively in [43]); even though a role for the ghrelin system in CRC pathogenesis has been described in vitro and animal studies, its relationship in human trials has been less conclusive. In this sense, some studies have shown that lower circulating ghrelin levels are associated with an increased risk of CRC, whereas others have reported no significant correlations or even an inverse correlation. Overall, it seems that most likely, reduced levels of ghrelin promote a metabolic proinflammatory environment that can be conducive to the development and growth of tumors in the context of CRC [43].

##### Resistin

Resistin is produced by the stromovascular fraction of adipose tissue and peripheral blood monocytes. It is hypothesised that upregulation of resistin expression increases intracellular lipid content, which in turn is associated with obesity-related inflammation. Thus, high resistin levels could be contributing to the characteristic cancer inflammatory state, as has been described for breast cancer [59] and non-small cell lung cancer. Furthermore, it has been reported that high serum resistin levels correlate with tumor grade and poor prognosis in CRC patients [60], whereby resistin binds to Toll-like receptor 4 on the colon cancer cell membrane and promotes proinflammatory signalling pathways. However, its exact role in CRC has yet to be elucidated [37]. 

#### 3.1.2. Cytokines and CRC

The existing relationship between chronic inflammation and increased cancer incidence has been previously reviewed [61]. Cytokines released by tumor-infiltrating immune cells and tumor cells are thought to be one of the mediators implicated in the link between chronic inflammation and CRC pathogenesis. Interleukin-6 (IL-6) and C-reactive protein (CRP) are the most studied to date and have been associated with larger tumor size, metastasis, and mortality in CRC patients [62]. 

##### Interleukin-6

IL-6 has been shown to be involved in tumorigenesis, metastasis, and tumor-associated cachexia in various cancers [63] and is considered one of the most relevant cytokines in cancer progression [64]. Furthermore, IL-6 is overexpressed in CAAs, promoting epithelial-mesenchymal transition (EMT)-phenotype and radio-resistance in breast cancer [65,66]. Ex vivo co-culture of colon cancer cells with visceral adipocytes from obese subjects induces the production of proinflammatory chemokines including IL-8, MCP1, and IL-6. These results suggest that the crosstalk between adipocytes and CRC cells contributes to the inflammatory milieu of the CRC TME [15]. On the other hand, IL-6-deficient mice have shown decreased tumor growth and increased tumor-infiltrating immune cells in the TME [67]. Moreover, increased pre-operative levels of serum IL-6 in CRC patients have been found to be predictive of prognosis [64] and associated with two to four times higher risks of overall mortality [63,68,69,70]; furthermore, elevated post-treatment circulating levels have also been linked to a higher risk of all-cause mortality over a 10-year follow-up period [62]. 

##### C-Reactive Protein

CRP is a well-recognized hallmark of inflammation and has been shown to be an adequate inflammation-based and feasible prognostic marker for CRC, whereby increased plasma levels are associated with a worse prognosis for CRC [71,72], larger tumor size [63], higher recurrence rates [73], and decreased overall survival [63]. As with IL-6, elevated preoperative serum levels were found to be predictive of the malignant potential of the tumor in CRC patients [72,74]. Interestingly, it has been reported that proinflammatory biomarkers such as CRP correlate better with patient’s VAT than BMI; moreover, two independent studies reported no association between CRP levels and overall mortality and high BMI/waist circumference in CRC patients [62]. This highlights the fact that VAT, and not BMI, could be more predictive of CRC outcome.

Thus, even though the evidence so far points quite clearly to the fact that in a state of excess adipose tissue, there is an increase in inflammatory adipokines and cytokines, their implications in CRC still need to be fully teased out. By understanding their role in CRC progression, they could serve as a helpful tool in identifying those patients at a higher risk for disease progression and mortality. 

### 3.2. Adipocytes as an Integral Part of the TME: Cancer-Associated Adipocytes

As discussed previously, there is a molecular interaction between fat depots and cancer cells mediated by hormones, adipokines, and inflammatory cytokines, which can reach the TME after being released at the systemic level. Nonetheless, the molecular crosstalk between both tissues could be even more direct since the colon is in physical contact with two of the main visceral fat depots in the human body: the omentum, which is located in the anterior peritoneal cavity, and the mesentery, which connects the colon with the posterior abdominal wall. In this context, the tumor mass of more advanced tumors can eventually extend through the serous membrane to establish direct contact with these fat depots or even generate metastatic sites within them [12]. Moreover, submucosal fat deposition is commonly observed in patients with intestinal diseases such as inflammatory bowel disease and cancer; therefore, adipocytes can also be part of the TME at earlier stages of CRC development [75]. 

As a result, it is frequent to observe adipocytes associated with the invasive margin of the tumor and infiltrated within the tumor mass, raising the question of how this close interaction is associated with CRC progression and prognosis. Interestingly, the evaluation of pathological images of 862 CRC samples from the Cancer Genome Atlas (TCGA) has demonstrated that adipocyte tissue infiltration is associated with lower overall survival [76]. This result suggests that adipose tissue can promote CRC progression, although the mechanisms mediating this process are still unclear. A recent report described that IL-6 and HGF secreted by tumor neighbouring VAT induce the expression of the metastatic marker CD44v6 in CRC cells and the transition from an epithelial consensus molecular subtype (CMS2) towards a mesenchymal subtype (CMS4) [77]. Thus, crosstalk between TME and VAT can induce cancer cell reprogramming and affect CRC development. Nonetheless, it is important to point out that recent refinement of molecular subtyping by single-cell analysis has shown that CMS4 is not an intrinsic CMS subtype but rather a non-homogeneous group of tumors with increased fibrosis [78]. In this sense, it could be interpreted that the paracrine secretion of protumorigenic factors from VAT could be an important contributing factor to fibrosis within the TME. 

Another important characteristic of the adipose tissue in close contact with the CRC TME is the profound phenotypic changes observed in adipocyte cells. In non-pathological conditions, adipocytes are highly plastic cells that can undergo phenotypic and metabolic adaptation in response to microenvironmental alterations. These changes include a size increase in the status of overnutrition [79], transdifferentiation into myofibroblasts due to skin fibrosis [80], or dedifferentiation into preadipocyte-like cells in the mammary gland during lactation [81]. Consequently, these cells are not only involved in fat storage and metabolic functions but are essential players in tissue remodeling and fibrosis. In this context, it is not surprising that adipocytes undergo phenotypic adaptation due to the interaction with tumor cells and the TME. This process is especially evident in the matrix surrounding invasive breast cancer, in which CAAs acquire a fibroblast-like phenotype characterized by loss of differentiation markers and reduced size [43,82]. In addition, CAAs show increased secretion of protumorigenic adipokines, including leptin and resistin, and inflammatory cytokines and chemokines, such as IL6, IL1β, TNFα, chemokine (C-C motif) ligand 5 (CCL5), CCL2, and C-X-C Motif Chemokine Ligand 8 (CXCL8) [50,83,84,85]. This proinflammatory phenotype is also associated with an enhanced expression of the ECM remodeling proteins, such as the matrix metallopeptidase 11 (MMP11) and procollagen-lysine, 2-oxoglutarate 5-dioxygenase 2 (PLOD2), and finally, with the production of high levels of fibronectin and collagen I [82,86,87]. Therefore, CAAs can be considered in this state as adipocyte-derived fibroblasts, contributing to the desmoplastic reaction in the tumor. 

On the other hand, the molecular mechanisms that drive the reprogramming process of CAAs within the CRC TME are yet to be disentangled. Adipocytes differentiate from stromal-vascular precursor cells found in adipose tissue. These cells mature in a multi-step differentiation process that involves first an adipocyte lineage commitment step, mediated by Wnt and bone morphogenetic protein families, followed by a maturation step mediated by the adipogenic factors CCCAAT/Enhancer Binding Protein α, β, δ (C/EBPα,β,δ) and the peroxisome proliferator-activated receptor gamma (PPARγ) [88]. Nonetheless, CAAs typically lack markers associated with terminal differentiation and show a fibroblast morphology resembling precursor cells, suggesting that CAAs result from a dedifferentiation process towards a progenitor-like status. It has been observed that soluble factors derived from tumor cells, such as TNFα and TGB-β1, can induce ex vivo dedifferentiation of adipocytes due to repression of PPARγ and C/EBPα expression [89]. Moreover, canonical Wnt activation by Wnt3a can also induce PPARy repression and dedifferentiation of both 3T3-L1 and human adipocytes [90], and Notch signalling is enough to induce dedifferentiation in an in vivo mouse model [91]. These signalling pathways can be strongly activated within the TME [92,93,94] and, thus, represent suitable mechanisms for CAA reprogramming towards a pre-adipocyte state. On the other hand, TME signalling has also been associated with the EM remodelling and fibroblast morphology of CAAs. Indeed, it has been observed that adipocytes present in the human breast tumor invasive front are stimulated by cancer cells to express the matrix metalloproteinase 11 (MMP11), inducing adipocyte reprogramming and EM remodelling [89]. Additionally, tumor-released plasminogen activator inhibitor-1 (PAI-1) can induce PLOD2 expression in adipocytes, followed by collagen reorganization at the tumor-adipose periphery [86]. Thus, the crosstalk between adipocytes and cancer cells participates in the EC remodelling of the TME and in the process of connective tissue invasion, suggesting that it could constitute a suitable target for metastatic cancer.

#### Metabolic Functions of CAAs within the TME

In general, CAAs show fibroblastic morphology with a reduced number and size of lipid droplets [95] and promote a catabolic metabolism, leading to the release of high-energy metabolites in the TME, including lactate, pyruvate, and adenosine triphosphate [96]. Because tumoral cells are highly proliferative and energy-demanding, the uptake of these metabolites from the TME may promote tumor growth. In addition, CAAs are lipolytic, producing free fatty acids (FFA) from lipid droplets that can be taken up by cancer cells for fatty acid β-oxidation (FAO) [97]. The activation of FAO pathways has been associated with cancer proliferation and stemness and, therefore, the release of FAA in the TME from adipocytes is a potential protumorigenic mechanism [98]. However, it is a matter of debate how these lipids are transferred into cancer cells and their relevance in the context of obesity. It has been demonstrated that cancer cells can uptake FFA from the TME using specialized transporters, such as CD36 and the solute carrier protein family 27 (SLC27). Interestingly, the deletion of CD36 alone is sufficient to hinder tumor growth in prostate cancer cells, indicating that tumors depend on the exogenous uptake of FFA [99]. Nonetheless, recent reports indicate that extracellular vesicles (EVs) are also implicated in the transport of FFA and the activation of the FAO pathways. In this context, it has been shown that melanoma cells internalize EVs containing FAO-related proteins and substrates from adipocytes, promoting migration and invasion [100]. These fatty acids are accumulated in lipid droplets in the cancer cells and are then released by lipophagy and used for FAO. This process is heightened in obesity, providing a molecular link between obesity and melanoma progression [101]. Moreover, because EVs can transport biomolecules at the system level, naïve adipocytes (not dedifferentiated) located in fat depots, such as VAT, can potentially drive cancer progression. On the other hand, the uptake of FFA has also been observed in colon cancer cells, allowing them to survive nutrient deprivation in an ex vivo co-culture assay [102], suggesting that this process is a common hallmark of cancers associated with adipocyte-rich environments. Finally, colon cancer cells show increased lipid droplet numbers which are highly enriched in cyclooxigenase-2 (COX2) and prostaglandin E synthase 2 (PGES2) [103]. These results suggest that lipid bodies are actively producing eicosanoids, which may be involved in inflammation and immune functions in CRC.

## 4. Future Directions and Strategies

Even though CRC is considered to be one of the most preventable cancers, it is still one of the most common cancers due to the overall unhealthy lifestyle of the population. Most importantly, it is still one of the deadliest [1]. The fact that obesity and being overweight have been recognised as modifiable risk factors in many cancers, including CRC, has opened up the possibility of targeting prevention and treatment strategies from a different perspective [6], which has only been explored recently. It is known that obesity increases cancer risk but at the same time may improve prognosis and survival, referred to as the “obesity paradox” [104], which has been observed for a variety of cancers. However, obesity has been associated with greater mortality and poorer outcome in CRC patients [105,106], and therefore, in order to understand the clinical implications of overweight and/or weight loss pre- and post-diagnosis, more research is necessary to exclude the impact of confounders and biases which may be driving this association [107]. Thus, although it is still unclear whether weight loss per se in obese individuals reduces the risk of developing cancer and/or improves prognosis, weight-loss strategies may be effective in improving CRC risk, prognosis, and survival [105]. 

Most studies use measurements such as BMI to establish correlations, which can be somewhat imprecise if no further measurements are carried out, such as adiposity indexes, and therefore can misrepresent obesity-related dysfunction associated with cancer. Further research regarding the role of individual adipose tissue compartments in CRC patients may be more informative and offer improved prognosis; however, assessing the adiposity index of a patient in a routine manner is time- and resource-consuming, and may not be plausible in many clinical settings. Nevertheless, waist circumference has been proposed as a useful surrogate for VAT and central adiposity [108,109,110], which is still under-used in clinical settings and which could serve as a proxy for VAT in CRC prevention and treatment strategies. 

Furthermore, specific adipokines and cytokines have also been shown to be useful predictors of disease progression and outcome, as well as being active players in CRC development, and should thus be considered when designing cancer-specific therapies. For example, as previously discussed in this review, leptin may play a role in cancer cell proliferation and apoptosis. In this sense, leptin is able to target its own receptor, which is over-expressed in a range of cancers, including CRC, leading to enhanced drug delivery properties. One study showed the potential of a leptin-derived peptide formulation in decreasing tumor growth and improved survival in BALB/c mice bearing C26 colon carcinoma than those treated with the same, non-Ob-R specific formulation [111]. Thus, even though stability studies and proof of validity in human tumor models are still needed, the results presented here open the possibility of a more targeted and efficient therapy, which highlights the importance of the components that make up the TME.

On the other hand, adipocytes in the TME also present themselves as an interesting niche to be explored for CRC treatment. It is known that these cells contribute to key aspects of cancer development, including fibrosis and metastasis. Therefore, studies focusing on these various mechanisms will allow us to understand better how obesity drives cancer progression and identify potential molecular targets for future therapeutic approaches against cancers associated with overweight or obesity. In addition, CAAs can contribute to the oncoinflammatory milieu of the TME. Because proinflammatory cytokines can restore antigen priming and improve effector immune cell infiltration in the TME, enhancing the proinflammatory phenotype of CAAs could be used for tumor immunotherapy. Moreover, current studies show that mesenchymal stem/stromal cells (MSCs) are a suitable vehicle for cytokine delivery [112]. The frequent presence of CAAs in the invasive front of colorectal and breast cancer indicates that these cells have a marked tropism to TME. Therefore, CAAs could also be used as a delivery system for cancer therapeutics. 

Finally, it has been suggested that conventional dendritic cells (cDCs) in VAT contribute to the maintenance of immune homeostasis by delaying the onset of obesity-induced chronic inflammation [113]. Due to the critical role of DCs in tumor development and tumor immune surveillance [114], this process could have important implications for comprehending the molecular mechanisms linking CRC and obesity as well as for the development of new strategies for the prevention and treatment of obesity-related malignancies.

## 5. Conclusions

CRC management, treatment, and prognosis have drastically improved in the last decade. However, the number of people diagnosed with this cancer is increasing in younger populations due to a variety of factors, including a rise in obesity. CRC is associated with both non-modifiable and modifiable risk factors, most of which have been extensively studied. However, the TME, which includes adipocytes mainly from the VAT, has only recently been recognized as a key player in the etiology and development of cancer and could be playing an important role in both cancer progression in a direct manner (for example, tumor promotion) and by indirect mechanisms (proinflammatory activity). Thus, this review aims to highlight the need to consider the TME in order to design more specific and targeted therapies.

## Figures and Tables

**Figure 1 ijms-23-13128-f001:**
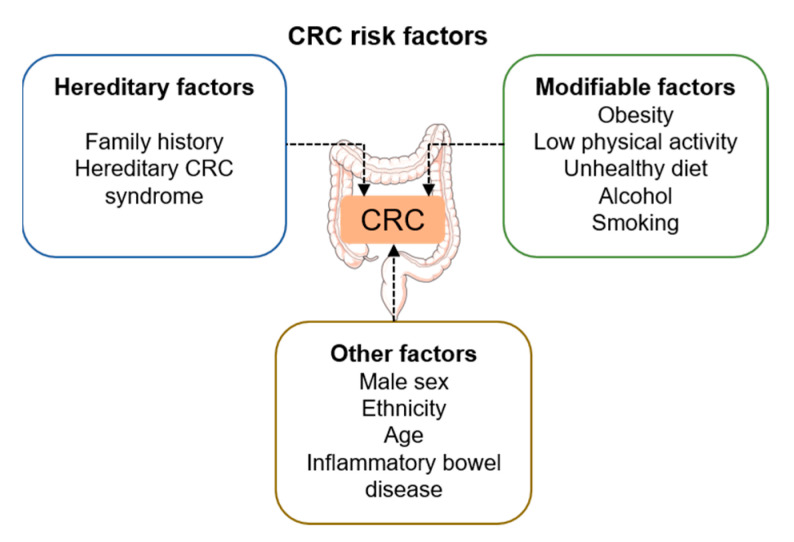
Overview of the main risk factors associated with CRC development.

**Figure 2 ijms-23-13128-f002:**
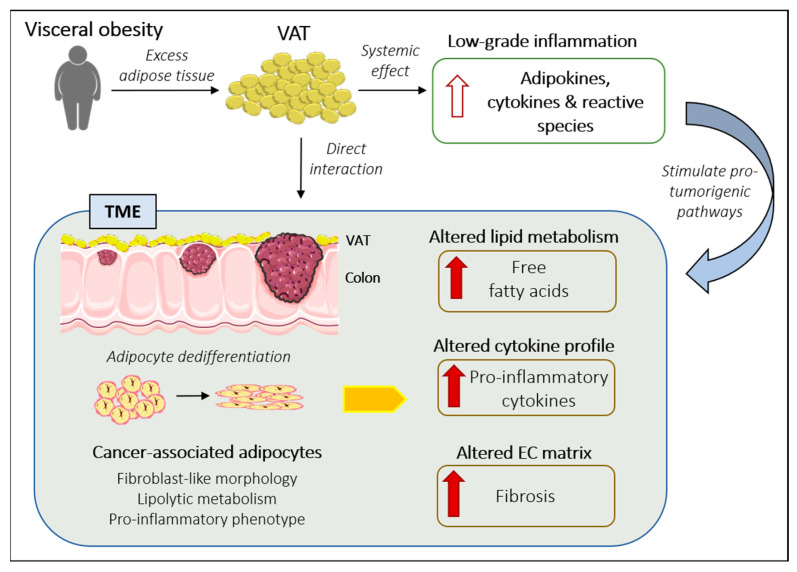
Summary figure which explains the potential relationship between VAT and CRC development and progression. EC: extracellular; TME: tumor microenvironment; VAT: visceral adipose tissue.

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
