# Peer review of "Insights behind the Relationship between Colorectal Cancer and Obesity: Is Visceral Adipose Tissue the Missing Link?"

_ijms, 2022, doi:10.3390/ijms232113128_

Round 1

Reviewer 1 Report

In the manuscript entitled ‘Insights behind the relationship between colon cancer and obesity: is visceral adipose tissue the missing link?’ Chaplin et al describe that visceral adipose tissue (VAT), a source of pro-inflammatory cytokines and systemic low-grade inflammation, may contribute to colorectal cancer (CRC) risk and development in individuals with obesity. In their review, Chaplin et al describe several cellular agents, such as adipokines and cytokines, that link VAT to the tumor microenvironment and thereby may influence CRC etiology and progression. The manuscript addresses a relevant issue, given the worldwide rise in obesity and early onset CRC. However, several areas require attention and should be addressed to warrant eligibility in IJMS.

Major points:

·         Throughout the whole manuscript, it is unclear if the authors refer to in vitro/in vivo/in patient studies. Please specify the type of evidence that is discussed.

·         The authors argue that the CRC patients are increasingly younger given the change in median age for diagnosis (72 in the early 2000’s VS 66 in 2020-2022). However, this can be due to the CRC screening implementation or screening uptake. Please elaborate.

·         One major point of concern is that important elements such as the microbiota, diabetes type 2, and diet are disregarded in this review. For example, food intake (such as more red meat and less fibers) change the microbiota and different microbiotas are associated to CRC. While an unhealthy diet (and a sedentary lifestyle) is the main cause of obesity, this major confounder is not discussed in the paper. Similarly, obesity and diabetes type 2 are related and diabetes type 2 is associated with a 1.3-fold increased risk of colorectal cancer. I wonder why the authors have not discussed this. Please take into account this major limitation or incorporate the relevant literature. See for instance: Peeters, P. J., Bazelier, M. T., Leufkens, H. G., de Vries, F., & De Bruin, M. L. (2015). The risk of colorectal cancer in patients with type 2 diabetes: associations with treatment stage and obesity. Diabetes Care, 38(3), 495-502.

·         Both leptin and adiponectin are related to insulin, and given the crucial role of insulin in obesity/diabetes type 2/metabolism of glucose into muscle and fat/etc. it is peculiar that the role of insulin is not discussed. Again, provide a thorough review of the relevant or a clear rationale on why insulin has been left out. See for example Farahani, H., Mahmoudi, T., Asadi, A., Nobakht, H., Dabiri, R., & Hamta, A. (2020). Insulin resistance and colorectal cancer risk: the role of elevated plasma resistin levels. Journal of gastrointestinal cancer, 51(2), 478-483.

·         Similar to the above stated points, inflammation seems to play a pivotal role in linking VAT to CRC. This is stressed by ghrelin, resistin, IL-6 and their inflammatory roles. However, the relation between inflammation, VAT and CRC has not clearly been reviewed.

·         The authors state that VAT/adipocytes are an integral part of the TME and regulate carcinogenesis (and more) in CRC. However, a colorectal tumor does not develop in the fat but in the colon/rectum, therefore, it is in direct contact/interaction with VAT? I would suggest to elaborate on this issue further.

·         Although Figure 2 provides a nice overview of possible links between VAT and CRC, I would suggest several improvements. First, if the authors are unsure of the phenotypical shift of macrophages (indicated by the question mark) I would suggest to omit this entirely from the figure. Furthermore, the function of the TME is not solely to provide energy to the tumor. The TME also includes immune cells (T cells etc.) that try to eliminate the malignant cells. Third, the link between the VAT and the TME is unclear: there is a direct link from the VAT to the TME, the VAT is in the TME, please clarify. Fourth, the muscle around the colon is missing, before the tumor reaches the VAT it is already quite invasive, and therefore the link with tumorigenesis is unclear in this figure. Moreover, it is unclear how the different adipokines and cytokines are interacting with CRC and this could be nicely illustrated in a figure. Therefore, I would suggest adding a figure that illustrates these interactions, or that it is zoomed in in the current figure.

·         The authors refer to VAT as a suitable (bio) marker for, for instance, CRC risk over BMI. However, it is unclear how to quickly and easily calculate an individual’s VAT/SAT index or VAT in general (a major pro of the BMI).

·         The gender differences in VAT remain undiscussed, while the male gender has a larger risk to develop CRC. In line with this, there are several comparisons between breast cancer and CRC and the role of VAT. Please specify if and how VAT may illustrate the role of gender in CRC. 

·         Finally, the conclusion can be strengthened. A point that strikes me is that severely ill cancer patients often lose a lot of weight and supposedly VAT. However, this does not improve the cancer cure. Thus, what are the implications of your review for patients and current cancer care?

Minor points:

·         Line 18: Visceral with a capital letter

·         Line 33: low physical activity, overweight etc. are not environmental factors.

·   Line 67: “Excess adipose tissue promotes the secretion of adipokines, cytokines, and reactive species, which could be one of the key aspects in promoting tumorigenesis” please explain.

·         The introduction would be strengthened by adding basic literature on the TME.

·         Line 110: full stop is missing.

·         Figure 1: Race is hereditary. I would omit race from this figure, or do the authors mean ethnicity/SES? If so, please elaborate.

·         Line 126-130: this is true for all cancers, not only for CRC. Please specify or omit.

·         Line 162-164: “This alters metabolism significantly and actively contributes to a state of low-grade inflammation, creating a favorable environment for tumor development.” How?

·         Line 177-181: “More specifically, they secrete adipokines (e.g., leptin, adiponectin, resistin, ghrelin) and inflammatory cytokines (e.g., tumor necrosis factor-α (TNF-α), interleukin-1 (IL-1), IL-6, IL-12 and IL-23), among others, all of which are particularly relevant to CRC development, and contribute to the characteristic systemic low-grade inflammation associated to visceral obesity” How?

·         Line 225: please elaborate on the implications of this finding.

·         Line 405-407: please provide references for this statement or attenuate.

·         Line 411-413: how does the obesity paradox relate to immunotherapy treatment in CRC?

·         Amiri Darban, S. et al. Targeting the leptin receptor: To evaluate therapeutic efficacy and anti-tumor effects of Doxil, in vitro and in vivo in mice bearing C26 colon carcinoma tumor. Colloids Surf B Biointerfaces 164, 107–115 (2018): seems very relevant and the results should be discussed more in depth.  

·         Line 403-433: how does this relate to CRC?

Author Response

  • RESPONSE TO REVIEWERS

    We thank the reviewers for all the interesting points raised, which we consider have helped to improve our manuscript significantly. We have addressed all points to the relevant degree and have made comments below to clarify our responses/changes in manuscript. All changes in the manuscript are indicated in red.

    REVIEWER 1:

    In the manuscript entitled ‘Insights behind the relationship between colon cancer and obesity: is visceral adipose tissue the missing link?’ Chaplin et al describe that visceral adipose tissue (VAT), a source of pro-inflammatory cytokines and systemic low-grade inflammation, may contribute to colorectal cancer (CRC) risk and development in individuals with obesity. In their review, Chaplin et al describe several cellular agents, such as adipokines and cytokines, that link VAT to the tumor microenvironment and thereby may influence CRC etiology and progression. The manuscript addresses a relevant issue, given the worldwide rise in obesity and early onset CRC. However, several areas require attention and should be addressed to warrant eligibility in IJMS.

    Major points:

    • Throughout the whole manuscript, it is unclear if the authors refer to in vitro/in vivo/in patient studies. Please specify the type of evidence that is discussed.

    The manuscript has been revised and changes/minor edits have been made where appropriate.

    • The authors argue that the CRC patients are increasingly younger given the change in median age for diagnosis (72 in the early 2000’s VS 66 in 2020-2022). However, this can be due to the CRC screening implementation or screening uptake. Please elaborate.

    We have included more information on this aspect, whereby we discuss the findings of Siegel et al. Interestingly, the authors describe that although an increase in screening could be a relevant factor in the increase of cases <50 years old, they also point out that those which drive the trend are younger in age. Furthermore, they also indicate that there has been an increase by equal amount in both early and late stage cancer, which is not consistent with early screening programs (lines 60-66).

    • One major point of concern is that important elements such as the microbiota, diabetes type 2, and diet are disregarded in this review. For example, food intake (such as more red meat and less fibers) change the microbiota and different microbiotas are associated to CRC. While an unhealthy diet (and a sedentary lifestyle) is the main cause of obesity, this major confounder is not discussed in the paper. Similarly, obesity and diabetes type 2 are related and diabetes type 2 is associated with a 1.3-fold increased risk of colorectal cancer. I wonder why the authors have not discussed this. Please take into account this major limitation or incorporate the relevant literature. See for instance: Peeters, P. J., Bazelier, M. T., Leufkens, H. G., de Vries, F., & De Bruin, M. L. (2015). The risk of colorectal cancer in patients with type 2 diabetes: associations with treatment stage and obesity. Diabetes Care, 38(3), 495-502.

    We thank the reviewer for pointing this out. Indeed, CRC is associated to a range of important modifiable risk factors (diet, lifestyle, gut microbiota composition) and metabolic conditions (obesity, type 2 diabetes), which are all also associated to inflammatory profiles. Thus, the relationship is complex and clearly multi-factorial. However, we consider that this has been extensively reviewed previously (we have added appropriate references in the manuscript) and the aim of the present review was to delve into the relationship between the TME and VAT in the context of CRC in particular. We have improved the TME definition (as per a minor point suggestion too) (lines 82-92) in which we define the aim of this review, as well as added some relevant information on this topic in section 2 (lines 135-142) to address this point.

    • Both leptin and adiponectin are related to insulin, and given the crucial role of insulin in obesity/diabetes type 2/metabolism of glucose into muscle and fat/etc. it is peculiar that the role of insulin is not discussed. Again, provide a thorough review of the relevant or a clear rationale on why insulin has been left out. See for example Farahani, H., Mahmoudi, T., Asadi, A., Nobakht, H., Dabiri, R., & Hamta, A. (2020). Insulin resistance and colorectal cancer risk: the role of elevated plasma resistin levels. Journal of gastrointestinal cancer, 51(2), 478-483.

    We appreciate this observation from the reviewer. Although it is very true that insulin plays a key role in the pathogenesis of obesity and obesity-related morbidities, the aim of our review was to provide a brief insight into the role of the main adipokines and cytokines associated to CRC. Indeed, it is an extensive topic, whereby various reviews which address this topic exclusively have already been published (referenced in our review) and thus the effects of insulin are outside of the scope of this section of the review.

    • Similar to the above stated points, inflammation seems to play a pivotal role in linking VAT to CRC. This is stressed by ghrelin, resistin, IL-6 and their inflammatory roles. However, the relation between inflammation, VAT and CRC has not clearly been reviewed.

    We are aware that there is not a section per se on inflammation, VAT and CRC; however, we considered that for the purpose of this review it was more appropriate to discuss it throughout the manuscript in all sections, highlighting how VAT and TME can both act on inflammation at different levels and in different manners.

    • The authors state that VAT/adipocytes are an integral part of the TME and regulate carcinogenesis (and more) in CRC. However, a colorectal tumor does not develop in the fat but in the colon/rectum, therefore, it is in direct contact/interaction with VAT? I would suggest to elaborate on this issue further.

    As per the existing literature, it seems that VAT interacts with the colon/rectum via indirect mechanisms (release of pro-inflammatory cytokines, for example), but also in a more direct manner, due to their close physical proximity. Thus, although the tumor originates in the colon/rectum, adipocytes found in the TME of said tumor could be infiltrating and engage in tumorigenesis; or the tumor could expand and cross out into the outer layers and interact directly with VAT. We have done our best to discuss this at various points throughout the review, particularly in section 3.2.

    • Although Figure 2 provides a nice overview of possible links between VAT and CRC, I would suggest several improvements. First, if the authors are unsure of the phenotypical shift of macrophages (indicated by the question mark) I would suggest to omit this entirely from the figure. Furthermore, the function of the TME is not solely to provide energy to the tumor. The TME also includes immune cells (T cells etc.) that try to eliminate the malignant cells. Third, the link between the VAT and the TME is unclear: there is a direct link from the VAT to the TME, the VAT is in the TME, please clarify. Fourth, the muscle around the colon is missing, before the tumor reaches the VAT it is already quite invasive, and therefore the link with tumorigenesis is unclear in this figure. Moreover, it is unclear how the different adipokines and cytokines are interacting with CRC and this could be nicely illustrated in a figure. Therefore, I would suggest adding a figure that illustrates these interactions, or that it is zoomed in in the current figure.

    We agree with the reviewer that Figure 2 could be improved to provide a more precise overview of the links between VAT and CRC. The figure intended to convey that VAT can interact with CRC at the systemic level or via direct interaction with the tumor mass. With this aim, we have omitted the phenotypical shift of macrophages from the figure and summarized the role of CAAs in the TME. We hope these modifications will clarify the role of VAT in CRC for the reader.

    • The authors refer to VAT as a suitable (bio) marker for, for instance, CRC risk over BMI. However, it is unclear how to quickly and easily calculate an individual’s VAT/SAT index or VAT in general (a major pro of the BMI).

    Indeed, the BMI measurement has its advantages (easy, accessible, non-trained staff…), yet it also presents some limitations, since it does not necessarily reflect the true metabolic status of the individual. We are also aware that suggesting that patients’ VAT should be routinely analyzed (via CT-scans, for example) is not feasible in many clinical settings due to reduced resources, and thus calculating adiposity indexes may only be possible in certain populations. However, waist circumference is a well-recognized surrogate for central adiposity, which is still not routinely used in clinical prevention strategies and which could be helpful in CRC risk screening. We thank the reviewer for pointing this out and have added an explanation on this topic in section 4 (lines 470-475).

    • The gender differences in VAT remain undiscussed, while the male gender has a larger risk to develop CRC. In line with this, there are several comparisons between breast cancer and CRC and the role of VAT. Please specify if and how VAT may illustrate the role of gender in CRC. 

    We thank the reviewer for pointing this out and have thus added a brief explanation in section 2 (lines 118-122) whereby we mention the fact that although men have been largely associated to a higher risk of CRC development, this could be partly due to the fact that the male sex tends to present more visceral fat. We have not gone into further detail as it is outside of the scope of this article, in which we aimed to focus on the molecular mechanisms occurring between VAT and TME in a CRC setting.

    • Finally, the conclusion can be strengthened. A point that strikes me is that severely ill cancer patients often lose a lot of weight and supposedly VAT. However, this does not improve the cancer cure. Thus, what are the implications of your review for patients and current cancer care?

     We agree with the reviewer that this is an important point, known as the “obesity paradox”, whereby overweight and obesity may increase the risk of certain cancers, yet can also have a protective effect post-diagnosis/survival. However, weight loss in cancer is most frequently cachexia-related (muscle loss) and can be associated to other risk factors (frailty, smoking, etc.), and this review focused on the potential deleterious effect of VAT-associated adipocytes on tumor formation and progression. We have elaborated this further in section 4 (lines 457-462), and have reformulated the conclusion with the hope it clarifies the goal of this review.

    Minor points:

    • Line 18: Visceral with a capital letter.

    Corrected.

    • Line 33: low physical activity, overweight etc. are not environmental factors.

    We have changed it to “modifiable”.

    • Line 67: “Excess adipose tissue promotes the secretion of adipokines, cytokines, and reactive species, which could be one of the key aspects in promoting tumorigenesis” please explain.

    We have included a sentence to clarify that this is discussed further in the review.

    • The introduction would be strengthened by adding basic literature on the TME.

     We have included a brief paragraph at the end of the introduction (lines 82-92) to provide more background.

    • Line 110: full stop is missing.

    Corrected

    • Figure 1: Race is hereditary. I would omit race from this figure, or do the authors mean ethnicity/SES? If so, please elaborate.

    We have eliminated the word race and replaced it by ethnicity. We have not gone into detail on the relationship between ethnicity and CRC as it is beyond the scope of this review.

    • Line 126-130: this is true for all cancers, not only for CRC. Please specify or omit.

    We appreciate the suggestion; however, we consider specifying CRC in this sentence relevant in the context of our review.

    • Line 162-164: “This alters metabolism significantly and actively contributes to a state of low-grade inflammation, creating a favorable environment for tumor development.” How?

    We have made some minor edits to the sentence and added a reference to clarify our point.

    • Line 177-181: “More specifically, they secrete adipokines (e.g., leptin, adiponectin, resistin, ghrelin) and inflammatory cytokines (e.g., tumor necrosis factor-α (TNF-α), interleukin-1 (IL-1), IL-6, IL-12 and IL-23), among others, all of which are particularly relevant to CRC development, and contribute to the characteristic systemic low-grade inflammation associated to visceral obesity” How?

    We discuss this in the adipokine/cytokine section and have also provided relevant references; however, we chose not to go into detail since it is a topic that has been previously reviewed.

    • Line 225: please elaborate on the implications of this finding.

    We have edited the sentence slightly to include more information.

    • Line 405-407: please provide references for this statement or attenuate.

    An appropriate reference has been added.

    • Line 411-413: how does the obesity paradox relate to immunotherapy treatment in CRC?

    We have rephrased the sentence on the obesity paradox, as per previous suggestion, and have edited the wording.

    • Amiri Darban, S. et al. Targeting the leptin receptor: To evaluate therapeutic efficacy and anti-tumor effects of Doxil, in vitro and in vivo in mice bearing C26 colon carcinoma tumor. Colloids Surf B Biointerfaces 164, 107–115 (2018): seems very relevant and the results should be discussed more in depth.
    • As suggested, we have discussed this paper further.
    • Line 403-433: how does this relate to CRC?

     Since we have rephrased some points in this section we hope our message is clearer.

Reviewer 2 Report

The authors provided a nice and concise overview on the plausible role of visceral adipose tissue for Colorectal Cancer risk and development. The references used include several recent studies. I have some minor comments and edit suggestions:

1. Row 18 "visceral" needs capitalization

2. Section 2. Should Lynch syndrome be mentioned?

3. Row 225, should be AdipoR2?

4. Row 242. 'Orexigenic' is a rarely used word, replace with something better known. 

5. Row 274. Since authors are trying to distinguish Adipokines (3.1.1) from Cytokines (3.1.2), use another word here for IL8/MCP1/IL-6 (cytokines, adipocytokines, chemokines?)

6. Row 289, use 'better' instead of 'best'?

7. Reference 24. Check the Author last and first names. Same for references 28, 44, 49 and 50. Check that the referencing style is consistent and according to the journal instructions.

Author Response

RESPONSE TO REVIEWERS

We thank the reviewers for all the interesting points raised, which we consider have helped to improve our manuscript significantly. We have addressed all points to the relevant degree and have made comments below to clarify our responses/changes in manuscript. All changes in the manuscript are indicated in red.

REVIEWER 2

The authors provided a nice and concise overview on the plausible role of visceral adipose tissue for Colorectal Cancer risk and development. The references used include several recent studies. I have some minor comments and edit suggestions:

  1. Row 18 "visceral" needs capitalization.

Corrected.

  1. Section 2. Should Lynch syndrome be mentioned?

It has been added, we thank the reviewer for pointing it out.

  1. Row 225, should be AdipoR2?

Corrected.

  1. Row 242. 'Orexigenic' is a rarely used word, replace with something better known. 

Instead of replacing orexigenic, we have added another sentence whereby we specify that ghrelin is the “hunger hormone” due to its stimulatory effect on food intake.

  1. Row 274. Since authors are trying to distinguish Adipokines (3.1.1) from Cytokines (3.1.2), use another word here for IL8/MCP1/IL-6 (cytokines, adipocytokines, chemokines?).

We have made the change as suggested.

  1. Row 289, use 'better' instead of 'best'?

Corrected.

  1. Reference 24. Check the Author last and first names. Same for references 28, 44, 49 and 50. Check that the referencing style is consistent and according to the journal instructions.

Thank you for pointing it out, we have revised all references.

Round 2

Reviewer 1 Report

Thank you for your revisions, I have no further comments.